# Study on the Matching of Surface Texture Parameters and Processing Parameters of Coated Cemented Carbide Tools

**Haochuan Yang, Shucai Yang * and Xin Tong**

Key Laboratory of Advanced Manufacturing and Intelligent Technology, Ministry of Education, Harbin University of Science and Technology, Harbin 150080, China
* Correspondence: yangshucai@hrbust.edu.cn

**Abstract:** Placing micro-textures on a tool surface can play an anti-wear and friction-reducing role and capture impurities and improve the tool-chip friction state, thus improving the cutting performance of the tool and the quality of the workpiece. To ensure the processing quality in the micro-texture-coated tool-cutting process, the process parameters and micro-texture parameters are limited to smaller parameters, which reduces the processing efficiency and increases the cost. Aiming at this problem, this paper designs orthogonal experiments of the cutting process and micro-texture parameters, builds an experimental platform for milling titanium alloy with a micro-texture-coated ball-end milling cutter, analyzes the influence of cutting parameters on tool milling performance and workpiece quality, establishes a high fitting prediction model, and optimizes parameters. The results show that the cutting parameters significantly affect the milling force, tool wear, and workpiece surface roughness, which are in the first response level, and the micro-texture parameters, which are in the second response level. It is proven that micro-texture has anti-wear and anti-friction effects, and it is found that micro-texture parameters affect the evaluation index by changing the distribution state of the micro-texture. It is found that the multiple linear regression model fits better. Parameter optimization results are: $v$ = 159.4232 (m/min), $a_p$ = 0.211 (mm), $f$ = 0.06 (mm/r), micro-pit diameter $D$ = 62.3429 (μm), distance from blade $L$ = 121.5184 (μm), and micro-pit spacing $L_1$ = 235.6443 (μm). It provides some guidance for the selection of micro-texture parameters and cutting parameters on a micro-texture-coated tool.

**Keywords:** coated cemented carbide tool; micro-texture; cutting parameters; matching

## 1. Introduction

As a lightweight metal, titanium alloy is often used in lightweight designs, and its material removal is large. However, because its processing quality is greatly affected by temperature changes, the current cutting of titanium alloy is mainly limited to smaller cutting parameters, which seriously affects the processing efficiency and indirectly increases the production cost [1,2]. At present, in order to ensure the machinability of titanium alloy, the tool surface will be coated to improve its friction and wear resistance. In addition, for the processing of difficult-to-machine materials such as titanium alloys, the application of surface micro-texture technology to the tool surface can improve the friction state during the cutting process, thereby reducing tool wear and extending tool life [3,4]. It can be found that the research on the matching of the cutting process and micro-textured tools is of great significance for the rapid realization of processing objectives and cost reduction, as well as the promotion of the continuous development of cutting to high efficiency and high quality.

Domestic and foreign scholars have studied the cutting process and micro-textured coated tools from different aspects. Zhou [5] studied the mechanism of tool wear under different cutting speeds through the combination of finite element simulation analysis and cutting experiment and analyzed the cutting parameters that affect the cutting performance index of the tool. Bi [6] used the control variable method to compare the experimental

and theoretical analysis of different cutting parameters to find the optimal combination of processing parameters. The experimental results showed that the surface roughness and milling trace of the workpiece was sensitive to low spindle speed and high feed speed, but not sensitive to cutting depth. Torrano [7] used three different finite element software programs to establish three-dimensional models of Inconel718 to analyze and predict residual stress. By changing the cutting speed and feed per tooth, the prediction model of residual stress was obtained, and the accuracy of the prediction model of different analysis software was compared. Li.A.H [8] found that when the cutting speed $v_c < 500$ m/min, the surface roughness decreases with the increase of cutting speed. When the cutting speed $v_c > 500$ m/min, the surface roughness value will increase with the increase in cutting speed. Garrido [9] prepared pit-shaped micro-textures with laser cladding processing technology and explored the relationship between micro-pit parameters and properties through tribological experiments. The results showed that changing the micro-texture parameters can change the performance of the micro-texture. Koshy [10] processed textures with different regular morphologies, such as grooves and micro-pits, on the rake face of the tooland carried out cutting experiments. The results showed that different micro-texture parameters have a great influence on the performance of the tool, and all micro-textures can improve the performance of the tool. Deba Kumar Sarma [11] et al. processed grooves and pits on the surface of coated cemented carbide tools, and analyzed the effect of micro-texture on coated tools, using cutting force, surface roughness and white layer thickness as evaluation indicators. It was found that, compared with a pure-coated tool, the addition of micro-texture made the tool performance better, while a micro-pittexture-coated tool was better than a micro-groove-coated tool in terms of cutting force, white layer thickness and friction coefficient. Tong X [12] designed an orthogonal dry-cutting test, established a micro-texture distribution model under different cutting depths, studied the influence of micro-texture parameters on tool milling performance under variable cutting parameters, and optimized the combination of texture parameters according to different evaluation indexes. To explore the role of micro-textured tools with different shapes in the process of cutting titanium alloys, Wang Liang [13] selected several micro-textures with different shapes and designed experiments with cutting parameters and micro-texture parameters as changing factors. The results show that the micro-texture shape had an effect on the positive pressure and surface friction coefficient of the tool surface, and the cooling effect was restricted by the lubrication method. Rodrigo et al. [14] studied the friction theory between the micro-textured surface of the tool and the chip. Experiments showed that when the chip was continuous, the friction between the rake face and the chip and the cutting force was reduced due to the micro-texture, and the effect of discontinuous chips was not obvious. Therefore, cutting parameters also affect the design of micro-texture. Cao Teng [15] designed surface micro-textures under different cutting parameters in combination with micro-texture simulation morphology to study the influence of surface micro-texture geometry on friction performance and provide a basis for the optimization of processing parameters of micro-textured parts. Cheng Li [16] found that, under the condition of variable cutting parameters, the placement of texture changes the contact performance between the tool chip and the workpiece, effectively reducing the main cutting force of the tool; after changing the cutting parameters, the cutting force of the textured tool was affected to varying degrees.

In summary, when a micro-textured coated tool is used to process the workpiece material, the cutting parameters have an important influence on the placement position of the micro-texture on the tool surface, the distribution form of the micro-texture, and the anti-wear and anti-friction effect of the micro-texture. Micro-texture parameters also play a key role. Therefore, this paper studies the change of tool milling performance under the joint influence of micro-texture parameters and cutting parameters when the coating parameters are fixed. Taking a micro-texture coated ball-end milling cutter as the research object, the experimental platform for milling titanium alloy was built. The influence rule and mechanism of the combined action of micro-texture parameters and

cutting parameters on milling force, rake face wear and workpiece surface roughness were analyzed. The regression prediction model with a high fitting degree was established for different evaluation indexes, and the parameters were optimized based on the artificial bee colony algorithm.

## 2. Material and Methods

### 2.1. Tools and Workpieces

A YG8 tungsten-cobalt ball-end milling cutter was selected for the test. The tool type was BNM-200, the diameter was 20 mm, the thickness was 5 mm, and the width was 15 mm. AlTiN coating was deposited on the surface of the tool by physical vapor deposition. The length of the special toolbar was 141 mm. The tool and its size are shown in Figure 1.

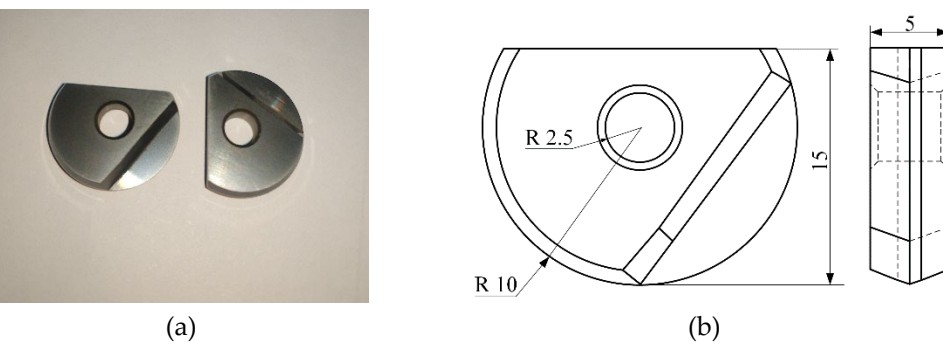

|     |     |
| --- | --- |
| (a) | (b) |

**Figure 1.** Tool and its size diagram. (**a**) Ball-end milling cutter; (**b**) Tool size diagram.

The workpiece material used in the test is Ti6Al4 V, and the workpiece size is $130 \times 76 \times 60$ mm. The titanium alloy square material test piece is shown in Figure 2.

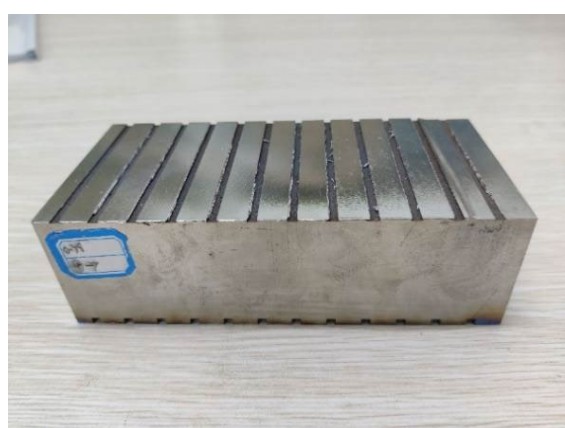

**Figure 2.** Test workpiece.

### 2.2. Orthogonal Experimental Design

Considering the cutting parameters (cutting speed, cutting depth, and feed rate) and micro-texture parameters (micro-pit diameter, distance from the edge, and micro-pit spacing), a total of 64 groups of six-factor eight-level orthogonal tests were designed. The orthogonal experimental design is shown in Table 1. The parameter diagram is shown in Figure 3.

### 2.3. Construction of the Experiment Platform

(1)    Tool micro-texture preparation

The ZTQ-50 fiber laser (Zhengtian Laser, Beijing, China) is used to prepare micro-texture on the rake face of the tool. The laser process parameters are: laser power, 40 W; number of scanning times, 7; and scanning speed, 1700 mm/s. After the micro-texture

preparation is completed, the tool surface is cleaned to ensure the quality of the tool micro-texture. The prepared micro-texture morphology was observed using an industrial camera tool image detection system, as shown in Figures 4 and 5. After measurement of the prepared micro-texture, the error of the dimension parameter of the micro-pit texture was less than 8%, within the allowable range, and the test results are considered credible.

**Table 1.** Orthogonal test parameters.

| Factor | Level | Cutting Speed $v$ (m/min) | Cutting Depth $a_p$ (mm) | Feed Rate $f$ (mm/r) | Micro-Pit Diameter $D$ (µm) | Distance from Blade $L$ (µm) | Micro-Pit Spacing $L_1$ (µm) |
|---|---|---|---|---|---|---|---|
| | 1 | 110 | 0.2 | 0.05 | 30 | 90 | 120 |
| | 2 | 120 | 0.25 | 0.06 | 40 | 100 | 140 |
| | 3 | 130 | 0.3 | 0.07 | 50 | 110 | 160 |
| | 4 | 140 | 0.35 | 0.08 | 60 | 120 | 180 |
| | 5 | 150 | 0.4 | 0.09 | 70 | 130 | 200 |
| | 6 | 160 | 0.45 | 0.1 | 80 | 140 | 220 |
| | 7 | 170 | 0.5 | 0.11 | 90 | 150 | 240 |
| | 8 | 180 | 0.55 | 0.12 | 100 | 160 | 260 |

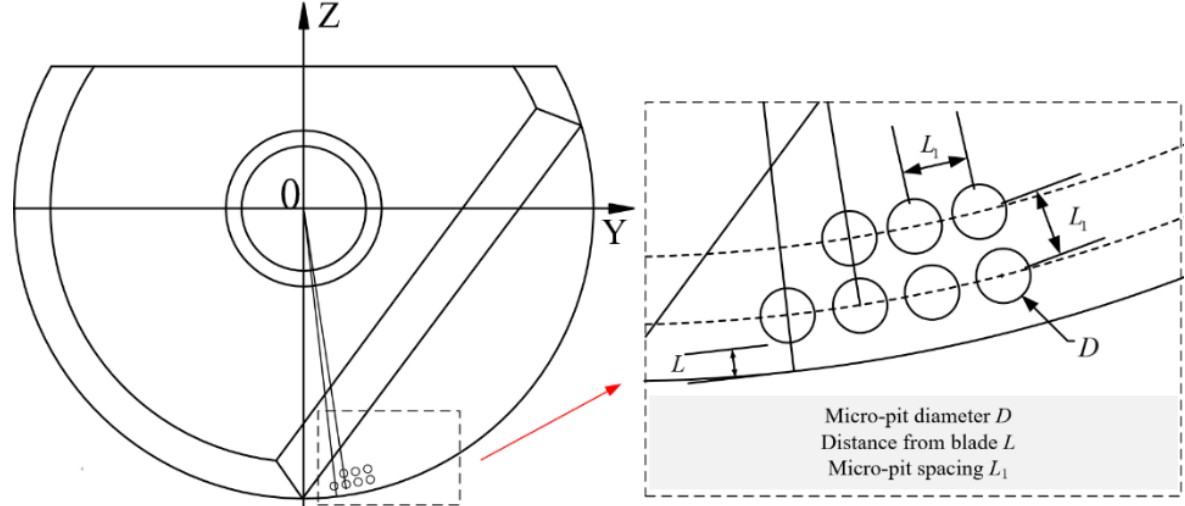

**Figure 3.** Parameter declaration.

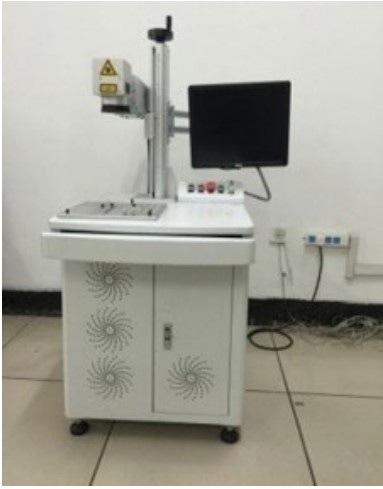

**Figure 4.** Micro-texture processing equipment.

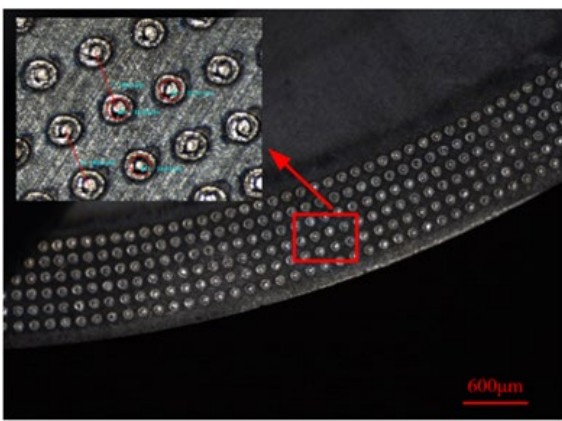

**Figure 5.** Texture parameter detection diagram.

(2)    Milling experimental platform

The milling method was one-way cutting along the milling. To protect the tool tip and give the tool a good cutting performance, the workpiece was placed at 15° with the horizontal plane. The ball-end milling cutter milling on the workpiece surface is shown in Figure 6. The cutting environment is dry cutting.

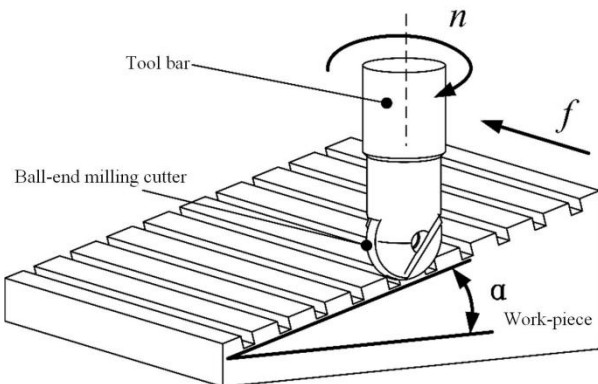

**Figure 6.** Milling diagram.

A three-axis vertical milling machine (VDL-1000E produced by Dalian Machine Tool Factory, Dalian, China) was used in the milling test. The clamping method of the titanium alloy workpiece is shown in Figure 7, and the angle between the surface to be machined and the horizontal plane of the machine tool guide rail is consistent with that of Figure 6.

(3)    Detection equipment

A rotary dynamometer (Kistler, Wintertour, Switzerland) was used to collect the milling force in the milling process. The sampling time was 60 s and the sampling frequency was 10,000 Hz. After removing the outliers, the extracted milling force diagram is shown in Figure 8. According to the image data, the component forces in the X, Y and Z directions were extracted, and the resultant force was calculated as the milling force in the milling process.

A white light interferometer (Taylor Hobson, Leicester, Britain) was used to measure the surface roughness Ra of the workpiece. Each specimen was measured at three points and the average value was recorded. The tool image detection system SH-VS4K and 4K fixed-multiple coaxial white light lens were used to photograph the wear indication of the rake face of the tool after milling, and then the picture was imported into Image-Pro software to measure the wear of the rake face. The wear of the rake face refers to the wear length perpendicular to the direction of the cutting edge. Taking the average value of the

wear amounts of the two rake faces, the test site and the Image-Pro interface are shown in Figure 9. Wear and roughness were measured at the same cutting stroke.

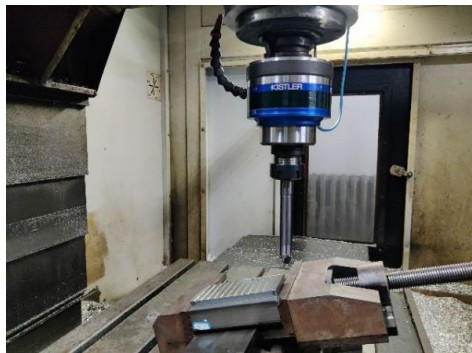

**Figure 7.** Workpiece clamping.

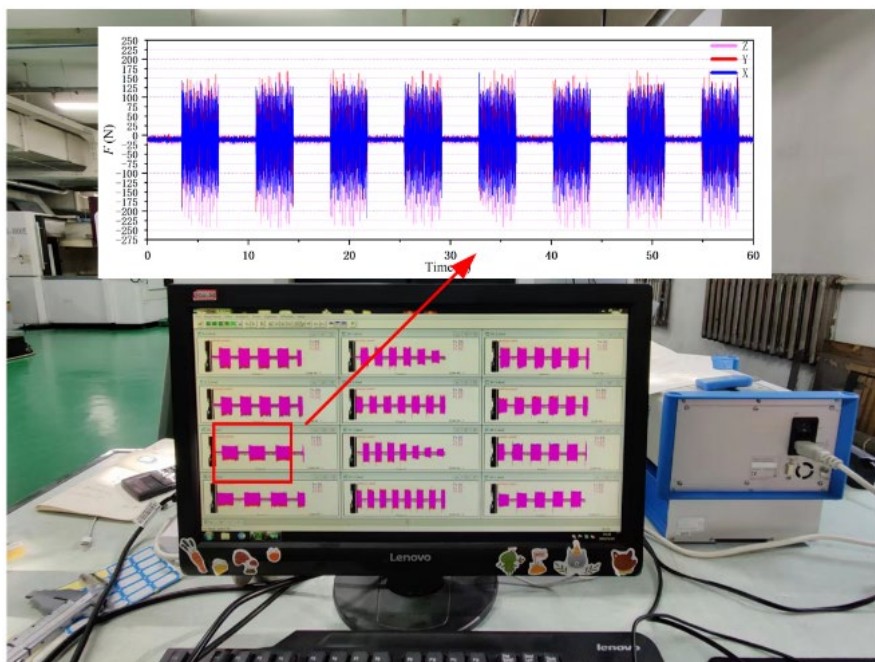

**Figure 8.** Milling force diagram.

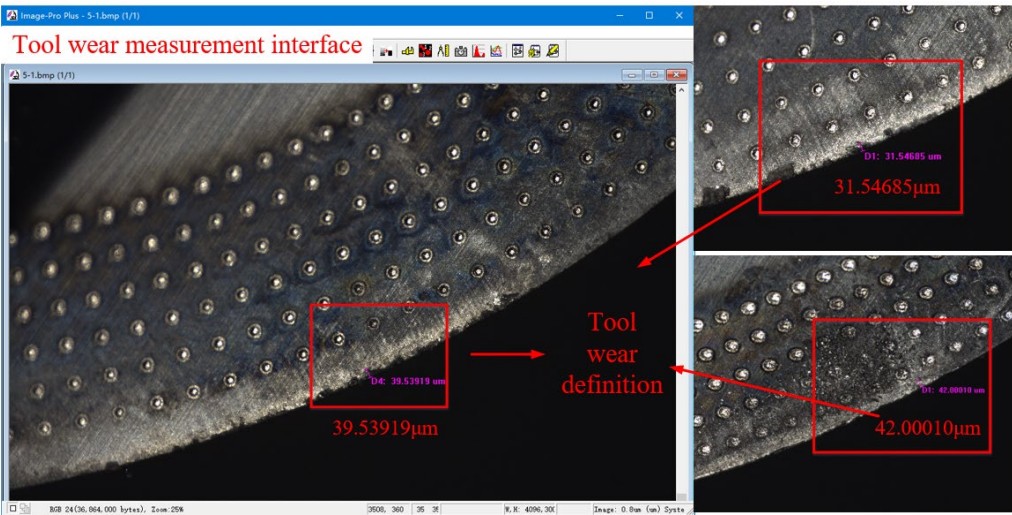

**Figure 9.** Tool rake face wear measurement.

## 3. Results and Discussion

### 3.1. Result Analysis of Milling Force

The experimental results of the milling force are shown in Table 2. The results of the range analysis are shown in Figure 10. According to the chart, among the factors affecting the milling force, the cutting parameters are in the first level, and the micro-texture parameters are in the second level.

**Table 2.** Experimental results of milling force (N).

| Number | Milling Force | Number | Milling Force | Number | Milling Force | Number | Milling Force |
|--------|---------------|--------|---------------|--------|---------------|--------|---------------|
| 1 | 209.28 | 17 | 220.81 | 33 | 208.92 | 49 | 250.33 |
| 2 | 236.3 | 18 | 224.18 | 34 | 273.11 | 50 | 228.78 |
| 3 | 286.72 | 19 | 215.69 | 35 | 273.17 | 51 | 263.1 |
| 4 | 269.36 | 20 | 272.83 | 36 | 292.2 | 52 | 269.22 |
| 5 | 322.27 | 21 | 293.44 | 37 | 318.69 | 53 | 290.22 |
| 6 | 341.57 | 22 | 322.42 | 38 | 332.68 | 54 | 326.84 |
| 7 | 331.21 | 23 | 365.27 | 39 | 342.79 | 55 | 342.48 |
| 8 | 363.51 | 24 | 349.07 | 40 | 369.44 | 56 | 398.74 |
| 9 | 237.58 | 25 | 215.2 | 41 | 215.92 | 57 | 260.12 |
| 10 | 219.94 | 26 | 258.37 | 42 | 226.7 | 58 | 255.24 |
| 11 | 274.51 | 27 | 259.07 | 43 | 298.02 | 59 | 259.99 |
| 12 | 292.63 | 28 | 319.36 | 44 | 310.61 | 60 | 299.1 |
| 13 | 296.87 | 29 | 336.37 | 45 | 316.33 | 61 | 298.87 |
| 14 | 321.46 | 30 | 294.65 | 46 | 323.95 | 62 | 335.45 |
| 15 | 342.73 | 31 | 345.83 | 47 | 334.53 | 63 | 383.75 |
| 16 | 293.33 | 32 | 396.02 | 48 | 356.82 | 64 | 401.74 |

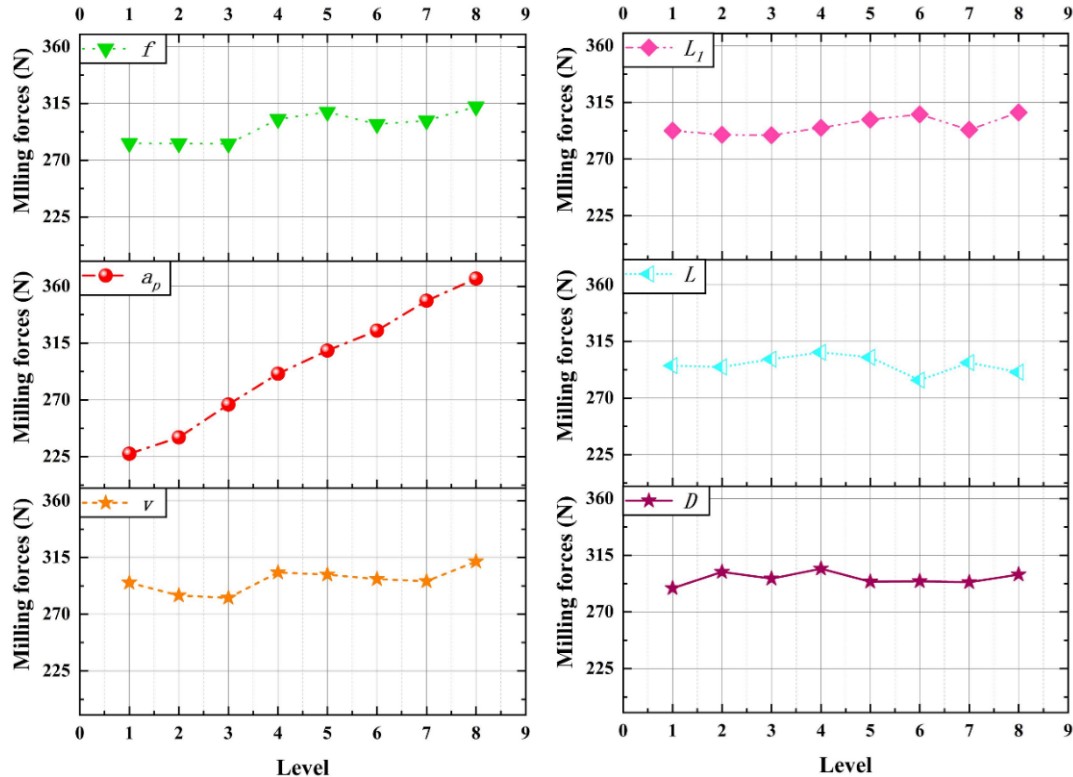

**Figure 10.** Milling force range analysis results.

It can be seen from the figure that, considering the cutting parameters, the cutting depth $a_p$ affects the milling force most significantly. The increase in cutting depth causes

the milling force to increase greatly. This is because the increase in cutting depth will increase the cutting width and the cross-sectional area of the cutting layer, increase the cutting deformation and friction, and increase the cutting force. Theoretically, when $a_p$ sad is doubled, the cutting force is also doubled. According to the experimental results, it is found that the milling force is increased by about 40%, which proves the anti-wear and anti-friction effect of micro-texture. The feed rate mainly affects the cross-sectional area of the cutting layer, thus affecting the milling force. The cutting speed has little effect on the milling force, and the milling force is in a relatively stable state.

Considering the micro-texture parameters, the distance from the edge has the most significant effect on the milling force. With the increase of the distance from the edge, the milling force decreases first and then increases. The reason is that when the distance from the cutting edge is small, the micro-texture near the cutting edge is easy to bond and block, the friction resistance increases, and the strength near the cutting edge is affected, so the milling force is large. With the increase of the distance from the edge, the micro-texture plays an anti-wear and anti-friction role and acts as a chip breaker. The positive effect is greater than the negative effect, and the cutting deformation and friction are reduced, so the milling force is reduced. When the distance from the edge is large, the effect of micro-texture decreases, and the milling force increases. The micro-texture diameter and micro-texture spacing mainly affect the cutting force by affecting the tool-chip contact area.

### 3.2. Analysis of Tool Wear Results

The experimental results of tool wear are shown in Table 3, and the results of range analysis are shown in Figure 11. According to the chart, the main influencing factors affecting tool wear are axial cutting depth $a_p$, feed rate $f$, and micro-texture diameter $D$.

**Table 3.** Experimental results of tool wear (μm).

| Number | Tool Wear | Number | Tool Wear | Number | Tool Wear | Number | Tool Wear |
|---|---|---|---|---|---|---|---|
| 1 | 46.21 | 17 | 61.69 | 33 | 28.29 | 49 | 71.97 |
| 2 | 34.36 | 18 | 49.72 | 34 | 55.82 | 50 | 50.01 |
| 3 | 45.4 | 19 | 39.22 | 35 | 62.88 | 51 | 53.95 |
| 4 | 54.33 | 20 | 37.38 | 36 | 82.49 | 52 | 55.9 |
| 5 | 66.6 | 21 | 47.61 | 37 | 57.27 | 53 | 76.57 |
| 6 | 58.5 | 22 | 66.63 | 38 | 74.52 | 54 | 28.53 |
| 7 | 47.38 | 23 | 78.48 | 39 | 53.4 | 55 | 89.25 |
| 8 | 72.97 | 24 | 101.66 | 40 | 67.46 | 56 | 79.19 |
| 9 | 41.47 | 25 | 50.37 | 41 | 48.8 | 57 | 16.41 |
| 10 | 38.85 | 26 | 69.93 | 42 | 29.76 | 58 | 57.79 |
| 11 | 54.34 | 27 | 36.55 | 43 | 88.22 | 59 | 48.99 |
| 12 | 45.34 | 28 | 41.87 | 44 | 68.25 | 60 | 43.52 |
| 13 | 59.45 | 29 | 56.87 | 45 | 77.96 | 61 | 47.2 |
| 14 | 66.04 | 30 | 55.44 | 46 | 63.54 | 62 | 91.15 |
| 15 | 66.6 | 31 | 73.08 | 47 | 79.18 | 63 | 85.35 |
| 16 | 50.43 | 32 | 90.31 | 48 | 69.69 | 64 | 112.19 |

It can be seen from the diagram that the tool wear VB increases with the increase in the three elements of cutting parameters in a certain range. The cutting depth $a_p$ has the greatest influence on the tool wear, which is slightly greater than the influence of the feed rate, and the cutting speed has less influence than the feed rate. The main reason is that, with the increase in feed rate, the cutting deformation work and friction work increase, and the increase in cutting heat makes the tool wear more severely. The increase in cutting depth increases the cutting heat, which is difficult to dissipate and aggravates the bonding phenomenon, so the cutting temperature increases. The increase in tool–chip contact area, the aggravation of the plowing phenomenon, and the increase in friction resistance lead to a significant increase in tool wear.

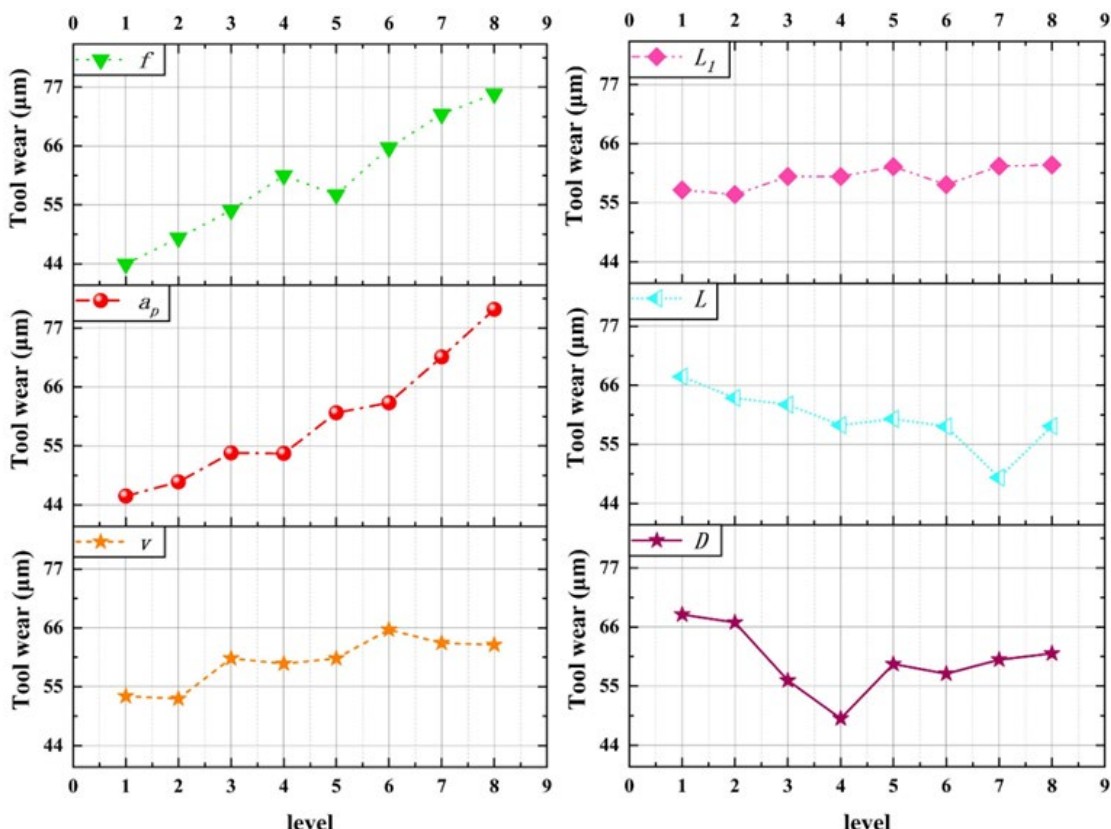

**Figure 11.** Tool wear range analysis results.

Considering the micro-texture parameters, the micro-texture diameter affects the tool wear most significantly. With the increase of micro-texture diameter, the tool wear decreases first and then increases. This is because when the diameter of the micro-texture is small, the volume of the micro-pit is small, and the micro-texture will be filled with impurities such as abrasive particles and wear debris in a short time, resulting in the blockage of the micro-texture, resulting in adhesion and increased tool wear. With the increase of the micro-texture diameter, the tool-chip contact area decreases, and the tool surface area increases, which is beneficial to heat dissipation, and the micro-pit edge is conducive to chip breaking, which reduces tool wear. When the micro-texture diameter is large, the edge of the micro-pit will produce a 'secondary cutting' phenomenon; the negative effect is greater than the positive effect, resulting in increased tool wear. The micro-texture spacing and the distance from the edge mainly affect the tool wear by changing the micro-texture distribution.

### 3.3. Result Analysis of Workpiece Surface Roughness

The experimental results of workpiece surface roughness are shown in Table 4, and the results of range analysis are shown in Figure 12. According to the chart, the main factors affecting the surface roughness of the workpiece are cutting speed $v$, feed rate $f$, micro-texture diameter $D$ and micro-texture spacing $L_1$.

It can be seen from the figure that, considering the cutting parameters, the cutting speed v affects the surface roughness of the workpiece most significantly. With the increase in cutting speed, the surface roughness of the workpiece decreases. This is because when the cutting speed is low, the cutting deformation is large, the system stability is poor, and burrs are prone to occur, resulting in poor surface roughness. When the cutting speed increases, the cutting deformation decreases, the milling force decreases, the system stability increases, and the surface roughness of the workpiece is improved. With the increase in feed rate, the surface roughness increases. This is due to the increase in feed rate, which increases the residual height and is prone to burr and vibration, resulting in poor surface roughness.

**Table 4.** Experimental results of workpiece surface roughness (nm).

| Number | Surface Roughness | Number | Surface Roughness | Number | Surface Roughness | Number | Surface Roughness |
|---|---|---|---|---|---|---|---|
| 1 | 343 | 17 | 349 | 33 | 322 | 49 | 371 |
| 2 | 305 | 18 | 338 | 34 | 302 | 50 | 301 |
| 3 | 373 | 19 | 336 | 35 | 326 | 51 | 312 |
| 4 | 394 | 20 | 300 | 36 | 363 | 52 | 260 |
| 5 | 376 | 21 | 354 | 37 | 334 | 53 | 323 |
| 6 | 399 | 22 | 328 | 38 | 361 | 54 | 227 |
| 7 | 399 | 23 | 371 | 39 | 339 | 55 | 300 |
| 8 | 374 | 24 | 402 | 40 | 312 | 56 | 316 |
| 9 | 319 | 25 | 328 | 41 | 300 | 57 | 296 |
| 10 | 318 | 26 | 377 | 42 | 272 | 58 | 319 |
| 11 | 376 | 27 | 328 | 43 | 322 | 59 | 244 |
| 12 | 370 | 28 | 332 | 44 | 297 | 60 | 297 |
| 13 | 359 | 29 | 288 | 45 | 354 | 61 | 252 |
| 14 | 370 | 30 | 317 | 46 | 320 | 62 | 329 |
| 15 | 341 | 31 | 379 | 47 | 329 | 63 | 256 |
| 16 | 366 | 32 | 343 | 48 | 294 | 64 | 345 |

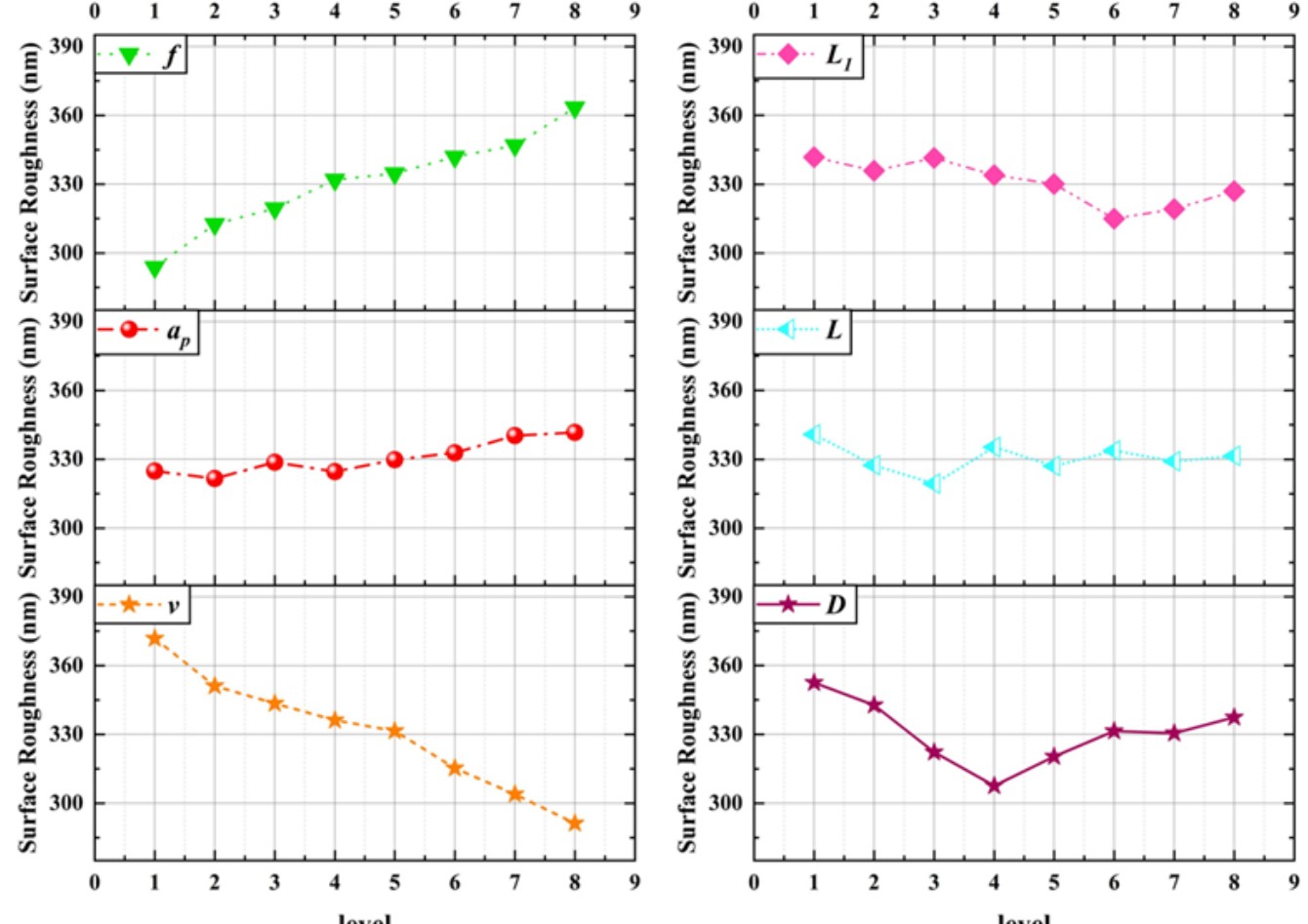

**Figure 12.** Analysis results of range of workpiece surface roughness.

Considering the micro-texture parameters, the micro-texture diameter affects the surface roughness of the workpiece most significantly. With the increase in micro-texture diameter, the surface roughness of the workpiece decreases first and then increases. This is because the change of micro-texture diameter will affect the tool-chip contact area,

the ability to store impurities such as abrasive particles and wear debris, and the heat dissipation area, thereby constraining the milling force, milling temperature and system stability, affecting the tool wear state, ultimately causing the surface roughness of the workpiece. The change of micro-texture spacing will affect the distribution density of micro-texture in the tool-chip contact area. A micro-texture density that is too small will weaken the micro-texture effect. If the density of the micro-texture is too large, the phenomenon of 'secondary cutting' at the edge of the micro-texture will be aggravated, and the positive effect of the micro-texture will be reduced.

## 4. Regression Model and Parameter Optimization

### 4.1. Establishment of the Prediction Model

(1) Empirical regression model

To clarify the complex exponential relationship between each test index and cutting parameters and micro-texture parameters, an empirical regression model was established, with milling force, tool wear and surface roughness as evaluation indexes. The established empirical model of milling force, surface roughness and tool wear of a micro-textured ball-end milling cutter for milling titanium alloy are:

$$
\begin{aligned}
F &= 267.9168 v^{0.1211} a_p^{0.4886} f^{0.1147} D^{0.0053} L^{-0.0194} L_1^{0.0668} \\
VB &= 1130.138 v^{0.2551} a_p^{0.5733} f^{0.6073} D^{-0.1} L^{-0.453} L_1^{0.1212} \\
Ra &= 10658.35 v^{-0.4792} a_p^{0.0503} f^{0.2171} D^{-0.0268} L^{0.016} L_1^{-0.0908}
\end{aligned} \tag{1}
$$

(2) Multiple linear regression model

In many practical problems, the dependent variable is usually affected by many factors. According to the empirical polynomial regression method, the predicted value of the regression equation may deviate greatly from the true value. It is of great significance to improve the prediction accuracy of the regression equation by retaining the items that have a significant influence on the evaluation index and establishing the 'optimal' regression model. The multiple linear regression method is developed based on this [17,18].

Finally, the equation established by the multiple regression principle is as follows:

$$
\begin{aligned}
F \ &= 296.65 + 9.04v + 71.13a_p + 14.43f + 0.7712D - 3.02L + 7.04L_1 \\
&\quad + 13.05va_p - 6.87vL_1 + 5.65a_p f - 6.82a_p L - 9.42fD
\end{aligned} \tag{2}
$$

$$
\begin{aligned}
VB \ &= 59.70 + 5.67v + 16.50a_p + 15.41f - 2.96D - 6.21L + 2.35L_1 \\
&\quad + 6.05va_p + 13.17vf - 10.81a_p D - 7.48fD + 7.25fL + 4.91LL_1
\end{aligned} \tag{3}
$$

$$
\begin{aligned}
Ra \ &= 330.42 - 38.14v + 9.93a_p + 30.03f - 2.44D + 1.33L - 10.93L_1 \\
&\quad - 33.41va_p + 11.01vL - 19.02a_p L_1 + 17.10DL + 22.88LL_1
\end{aligned} \tag{4}
$$

(3) BP neural network milling performance prediction model

Considering that there are many factors in this experiment, the mathematical equation established by the traditional multiple regression model have difficulty achieving the goal, so the milling performance is predicted by establishing a BP neural network. A BP neural network does not need to clarify the mapping relationship between independent variables and evaluation indicators. Through the training and operation rules of the neural network itself, it can output the most suitable training results under the neural network rules [19].

The training initial setting is: Sigmoid function as activation function, andeight hidden layer nodes. The experimental data are imported into Matlab. After the network structure is built, the experimental data are divided into 44 groups as the training set, and the remaining 20 groups as the test set. The output of the neural network prediction value is observed in the test set. Figures 13–15 show, respectively, the milling force, tool wear and surface roughness prediction value comparison chart and training process curve.

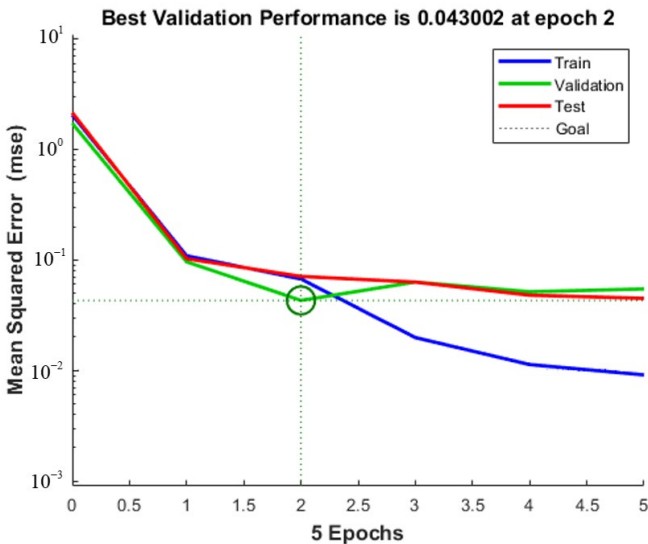

**Figure 13.** Milling force training process curve.

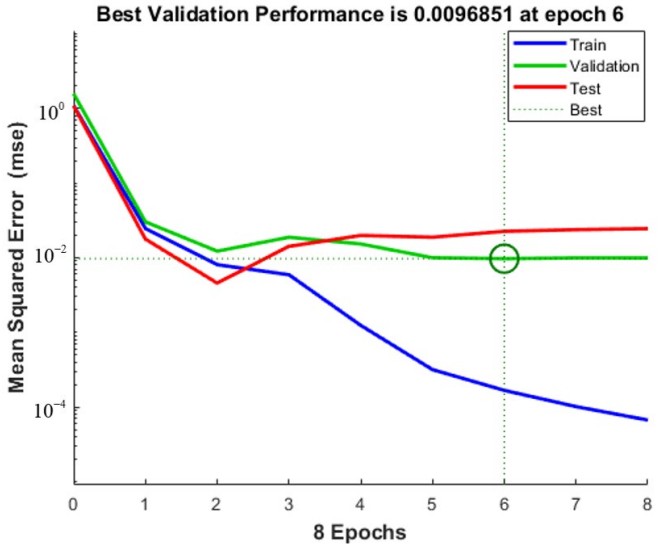

**Figure 14.** Tool wear training process curve.

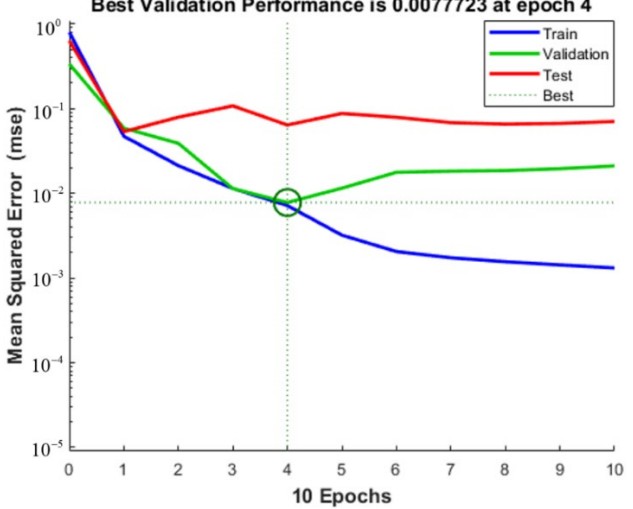

**Figure 15.** Surface roughness training process curve.

After the training is completed, the root means square error and determination coefficient of the neural network with the best training effect is obtained in Table 5. It can be seen that the coefficient of determination is above 75%, which proves that the training effect is good.

**Table 5.** Root mean square error and determination coefficient of neural network.

| Evaluating Indicator | Root Mean Square Error | Coefficient of Determination $r^2$ |
| --- | --- | --- |
| Milling force | 21.025 | 83.062 |
| Tool wear | 8.3378 | 76.211 |
| Surface roughness | 19.374 | 80.752 |

### 4.2. Significance Test and Optimization of the Prediction Model

To further investigate the accuracy of the prediction model, it is necessary to test its significance. The significance test of the established empirical model and multiple regression model is carried out to evaluate the model. When the significance test is passed, the root mean square error is used to describe the fitting degree between the predicted value and the actual value, and the model with a high fitting degree is selected to improve the prediction accuracy of milling performance.

(1)    Empirical model significance test

Through variance analysis, the results of the variance analysis of the empirical regression model are shown in Table 6.

**Table 6.** Variance analysis of empirical regression model.

| | | Degree of Freedom | Regression Sum of Squares | Mean Square | $F$ | Significance $F$ |
| --- | --- | --- | --- | --- | --- | --- |
| Milling force | Regression analysis | 6 | 0.3308 | 0.0551 | 77.0077 | $1.6395 \times 10^{-25}$ |
| | Residual error | 57 | 0.0408 | 0.0007 | —— | —— |
| | Grand total | 63 | 0.3716 | —— | —— | —— |
| Tool wear | Regression analysis | 6 | 0.9397 | 0.1566 | 19.2547 | $4.0847 \times 10^{-12}$ |
| | Residual error | 57 | 0.4636 | 0.0081 | —— | —— |
| | Grand total | 63 | 1.40343 | —— | —— | —— |
| Surface roughness | Regression analysis | 6 | 0.12848 | 0.02141 | 24.6066 | $3.6314 \times 10^{-14}$ |
| | Residual error | 57 | 0.04960 | 0.00087 | —— | —— |
| | Grand total | 63 | 0.17809 | —— | —— | —— |

According to the analysis of the variance table, it can be seen that the degree of freedom m between groups is 6, the degree of freedom within groups is 57, the total degree of freedom n is 63, and the significance level is 0.05. The milling force, rake face wear, and surface roughness P values of the empirical regression equation are $1.6395 \times 10^{-25}$, $4.0847 \times 10^{-12}$ and $3.6314 \times 10^{-14}$, respectively, which are far less than the significance level of 0.05. The F values of the three are 77.0077, 19.2547 and 24.6066, respectively. By checking the F-test critical value table, F (m, n-m-1) = F0.05 (6.56) = 2.266 can be obtained. The actual value of the statistic F is much larger than 2.266, and the significant difference can be judged. The regression models of milling force, rake face wear and surface roughness established by the empirical regression equations are highly significant.

(2)    Significance test of tbe multiple regression model

A significance test was performed on the regression equation, as shown in Table 7. According to the data results, it can be seen that the p-value of the multiple regression model of milling force F, tool wear VB and surface roughness Ra is far less than the significance level of 0.05, and the statistic F is also greater than 2.266. It can be judged that there is a

significant difference, and the multiple linear regression model is significant. In addition, the coefficient of determination ($R^2$) of the regression formula is close to 0.9, which is close to 1, indicating that the fitted multiple regression equation is better.

**Table 7.** Multiple regression equation milling performance significance.

| Evaluating Indicator | $R^2$ | Adjusted $R^2$ | $F$ | Significance $F$ |
|---|---|---|---|---|
| Milling force $F$ | 0.918 | 0.894 | 49.39 | <0.0001 |
| Abrasion loss $VB$ | 0.841 | 0.803 | 22.44 | <0.0001 |
| Surface roughness $Ra$ | 0.858 | 0.828 | 28.49 | <0.0001 |

(3)　Evaluation of the milling performance prediction model

The root mean square error is commonly used to describe the fitting degree between the measured value and the actual value. The smaller the root mean square error is, the higher the fitting degree is. It can be seen from Table 8 that the root mean square error of multiple regression analysis is smaller than that of the other two models, indicating that the model has a good fit, so the multiple regression model is better. Therefore, the multiple regression model is selected as the prediction model of milling performance and as the objective function of subsequent parameter optimization.

**Table 8.** Root mean square error results.

| Evaluating Indicator | Empirical Formula | Multiple Linear Regression | BP Neural Network |
|---|---|---|---|
| Milling force $F$ | 16.4745 | 15.9 | 21.025 |
| Abrasion loss $VB$ | 10.1494 | 7.69 | 8.3378 |
| Surface roughness $Ra$ | 21.4313 | 14.3 | 19.374 |

*4.3. Establishment of the Optimization Model of Micro-Texture Parameters and Cutting Parameters*

(1)　Objective function and constraint conditions

Since there are multiple optimization objectives, the data need to be dimensionless processed first, before establishing a model for an optimization. For the parameter optimization process, the objective function should be determined first. For the optimization problem of multiple objectives, the weighted objective function is often established by the integration method. The optimization objective function of micro-texture parameters and cutting parameters of the micro-texture ball-end milling cutter is as follows:

$$M'(X) = \lambda_1 F(X) + \lambda_2 VB(X) + \lambda_3 Ra(X) \tag{5}$$

In the formula, $\lambda_1$, $\lambda_2$, and $\lambda_3$ are weighted coefficients, which reflect the importance of the objective function, and X is the optimal design variable. Then the constraint conditions are established according to the selection of parameters in the actual test. The feed per tooth satisfies the constraint: 0.05 mm/r $\leq f \leq$ 0.12 mm/r; the cutting depth satisfies the constraint: 0.2 mm $\leq a_p \leq$ 0.55 mm; the cutting speed satisfies the constraints: 110 m/min $\leq v \leq$ 180 m/min; the micro-texture diameter satisfies the constraint: 30 µm $\leq D \leq$ 100 µm; the distance from the blade meets the constraint: 90 µm $\leq L \leq$ 160 µm; and the micro-texture spacing satisfies the constraint: 120 µm $\leq L_1 \leq$ 260 µm.

(2)　Implementation and verification of artificial bee colony algorithm parameter optimization model

The artificial bee colony algorithm is a kind of swarm intelligence optimization algorithm inspired by the behavior mechanism of bees picking honey [20]. Through Matlab programming, set the same weight of F, VB and Ra, and set relevant parameters and constraints to search for the optimal solution. When $v$ = 159.4232 (m/min), $a_p$ = 0.211 (mm),

f = 0.06 (mm/r), *D* = 62.3429 (μm), *L* = 121.5184 (μm), and *L₁* = 235.6443 (μm), the milling force, the surface roughness of the workpiece and the amount of tool wear are minimized. The search results are shown in Figure 16.

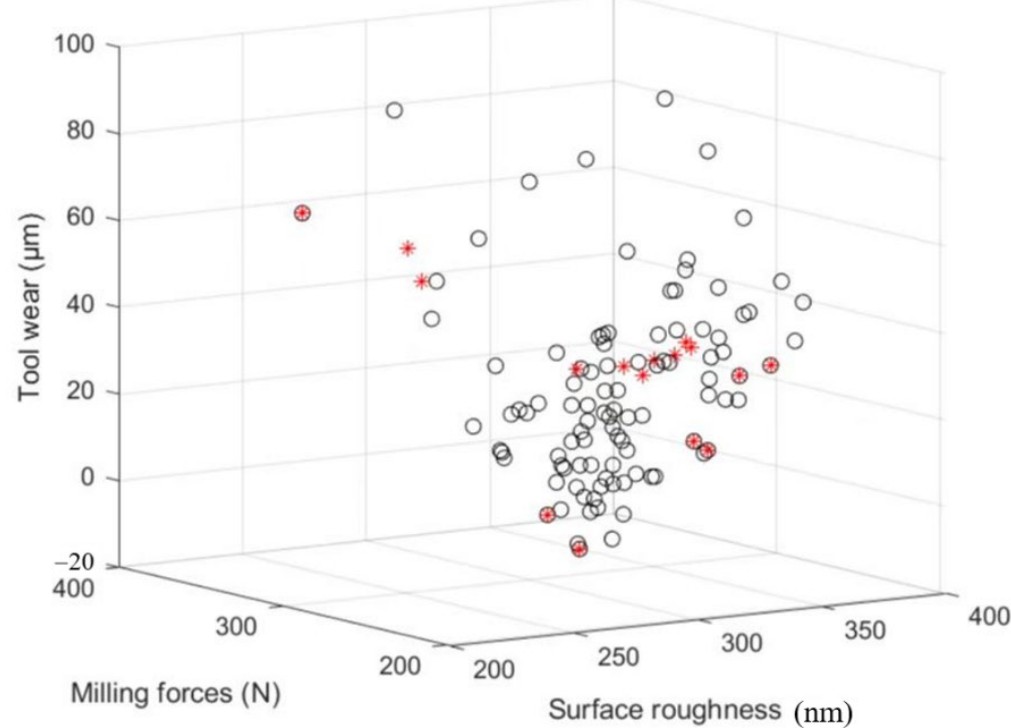

**Figure 16.** Artificial bee colony algorithm optimization results.

Based on the above optimization parameters, the verification test of the ball-end cutter milling titanium alloy is carried out. The milling force F, tool wear VB and roughness Ra in the machining process are measured, and the error between the optimization value and the experimental value is recorded. The optimization and experimental results are shown in Table 9.

**Table 9.** Relative error of experimental results.

| Evaluating Indicator | Optimization | Actual Measurement | Relative Error |
|---|---|---|---|
| Milling force (N) | 203.58 | 219.52 | 7.26% |
| Tool wear (μm) | 14.89 | 16.47 | 9.59% |
| Surface roughness (nm) | 245.87 | 263.71 | 6.77% |

According to Table 8, the relative error between the optimization results and the experimental results is within 10%, which verifies the accuracy of the established optimization model.

## 5. Conclusions

1.  The orthogonal experiments of cutting parameters and micro-texture parameters are designed, and an experimental platform for milling titanium alloy with micro-texture coated ball end milling cutters is established. The results show that the cutting parameters affect the milling force, tool wear and workpiece surface roughness more significantly, and the micro-texture parameters are in the second response level. It is found that micro-texture parameters limit the effect of micro-texture by changing its distribution in the insertion region, thereby affecting milling performance and workpiece surface quality. The effect of micro-texture on anti-wear, friction reduction, heat dissipation and chip storage has been demonstrated.

2. Based on the empirical regression method, multiple linear regression method and BP neural network method, the prediction models of milling force, tool wear and workpiece surface roughness are established. The root mean square error is used to describe the fitting degree, and it is found that the multiple linear regression method has the highest fitting degree.

3. Taking milling force, tool wear and workpiece surface roughness as evaluation indexes, the cutting parameters and micro-texture parameters are optimized based on the artificial bee colony algorithm. The optimization results are: $v = 159.4232$ (m/min), $a_p = 0.211$ (mm), $f = 0.06$ (mm/r), $D = 62.3429$ (μm), $L = 121.5184$ (μm), and $L_1 = 235.6443$ (μm).

**Author Contributions:** Conceptualization, S.Y. and H.Y.; validation, H.Y.; formal analysis, S.Y.; writing—original draft preparation, H.Y.; writing—review and editing, S.Y.; funding acquisition, S.Y. and X.T. All authors have read and agreed to the published version of the manuscript.

**Funding:** This research was funded by the National Natural Science Foundation of China, grant no. 51875144 and 52005140.

**Institutional Review Board Statement:** Not applicable.

**Informed Consent Statement:** Not applicable.

**Data Availability Statement:** Not applicable.

**Conflicts of Interest:** The authors declare no conflict of interest.

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
