# Peer review of "Study on the Matching of Surface Texture Parameters and Processing Parameters of Coated Cemented Carbide Tools"

_coatings, doi:10.3390/coatings13040681_

Round 1

Reviewer 1 Report

Notes in the attachment.

Reviewer 2 Report

I have some questions:

1. Have the authors considered in carrying out this experiment with other kind of cemented carbides as susbtrate such as WC-Co-Ni, WC-Co-Fe-Ni or even WC-TiC-(Ta,Nb)C-Co?

2. Have the authors considered in working with other type of coatings instead of using AlTiN?

3. Have the authors considered in using other workpieces instead of Ti6Al4V?

Author Response

Response to reviewers

Dear editors,

Thank you very much for your letter and the comments from the reviewers about our paper submitted to the coatings-2282142 entitled "Study on the matching of surface texture parameters and processing parameters of coated cemented carbide tools". We have checked the manuscript and revised it according to the comments. In the revised manuscript, the modified parts are highlighted with different colors. We submit here the response to the editor and reviewers' comments in details:

We really appreciate the editor and the reviewers' valuable suggestions. To address the reviewers' comments, substantial changes have been made to the manuscripts. For each of the reviewer's comment, we have provided detailed responses and made corresponding revisions in the manuscript. We believe that we have addressed all the issues raised by the reviewers. Thanks again for the valuable work on this manuscript.

Reviewer(s)’ Comments to Author:

*********************************************************************

Reviewer 2:

*********************************************************************

I have some questions:

  1. Have the authors considered in carrying out this experiment with other kind of cemented carbides as susbtrate such as WC-Co-Ni, WC-Co-Fe-Ni or even WC-TiC-(Ta,Nb)C-Co?

RESPONSE

In the team's early research, the better tool material for cutting titanium alloy has been determined, and other tool materials will not be considered.

  1. Have the authors considered in working with other type of coatings instead of using AlTiN?

RESPONSE

Thank you very much for your question. AlCrN, AlSiTiN and other coatings will be considered in the team's later research.

  1. Have the authors considered in using other workpieces instead of Ti6Al4V?

RESPONSE

Due to the advantages of high strength, good corrosion resistance and low temperature performance, titanium alloy has been widely used in aerospace, shipbuilding, medical devices and other fields. However, due to the problems of high chemical activity and low thermal conductivity elasticity of titanium alloy, it is easy to cause the problems of high cutting force and cutting temperature and serious tool wear during processing. Therefore, titanium alloy is used as workpiece material in this study. As for the model of titanium alloy, other types will be considered in our later research.

*********************************************************************

Reviewer 3 Report

A fairly large number of studies have been devoted to the study of the effect of microtexturing of the working surfaces of the tool on its cutting properties. I can recommend another interesting review on this topic: https://doi.org/10.1016/j.cirp.2021.05.006 .

Your article needs a very serious revision. I have quite a lot of comments.

1. The annotation needs to be redone. It is not clear from it why a tool with microtexture works better. You have a tool with replaceable plates, this is not reflected. Without reading the article, I will not understand what D, L and L1 are.

2. Error in line 123: "the scanning speed is 7 times".

3. Figures 3 and 6a are superfluous. They do not carry a semantic load.

4. According to Figure 6b: why was the instrument taken with such a large departure? To make the vibrations felt stronger?

5. When working with a microtextured tool, a lubricant is usually used to fill the craters of the texture. At the same time, the effect is much stronger. Have you been working dry?

6. According to Figure 7. Cutting modes are not presented. How was the value of the cutting force measured, presented in Table 2? What was the measurement error?

7. Figures 8,9,10 is superfluous. It is enough to describe the equipment used.

8. Figures 4 and 11 are almost identical. There is no scale with a size anywhere. The laser exposure zone is not visible. We need to analyze it using SEM.

9. Tables 2, 3, 4 are completely incomprehensible. What are the experiment numbers?

10. How was the wear of the plates measured? How were the wear criterion chosen? Did its character change depend on the parameters of the microtexture? Provide photos and their description.

11. The experimental error is not shown in Figures 13 and 14.

12. What is the roughness of 300 µm? Who needs such a thing? What did you measure? Provide an image.

13. Graphs 15a, 16a, 17a, 18, 19 have no physical meaning. Is it your dependence of the physical quantity on the experiment number? What to do about it? There is not a word about these experiments in the article.

Round 2

Reviewer 1 Report

I accept the authors correction in the article.

Author Response

Thank you very much for your comments. We will make necessary modifications to the paper based on your comments.

Reviewer 3 Report

Sirs, your article has become much better after correction, but I still have a few questions. As a wish: Using a lubricant, at least from an aerosol balloon, such as Molykote, would noticeably enhance the effect of texturing. Take a look https://doi.org/10.3390/coatings12121906.

According to the article:

1. Figures 5 and 9 are the same. By the way, it would be nice to crop Figure 9.

2. I don't understand what parameters the experiment has, for example, number 17? Or what is the difference between experiment number 5 and experiment number 39? Where can I find this data?

3. You have a level indicated on the graphs along the axis. What for? Specify the appropriate physical quantity. Specify the confidence interval corresponding to the parameter measurement error.

4. I fundamentally do not like the graphs 13a, 14a, 15a and in Figure 16. It is better to remove them. A table is enough to assess the adequacy of your model.

5. And yet what effect did you get, thanks to the texturing of the tool. You forgot to mention this in the conclusions.

Round 3

Reviewer 3 Report

There is very little left to fix.

1. Tables 2,3,4 in this form is absolutely unreadable. Maybe it's better to remove them? And the graphs in Figures 10, 11, 12 can be made larger.

2. In Figure 9, the labels are too small. It's impossible to read. The nature of wear is poorly visible. It won't be possible to make it bigger.
